# Lack of androgen receptor SUMOylation results in male infertility due to epididymal dysfunction

Fu-Ping Zhang[1,2], Marjo Malinen[3], Arfa Mehmood[4], Tiina Lehtiniemi[1], Tiina Jääskeläinen[5], Einari A. Niskanen[5], Hanna Korhonen[1], Asta Laiho[4], Laura L. Elo[4], Claes Ohlsson[6], Noora Kotaja[1], Matti Poutanen[1,2,6], Petra Sipilä[1,2] & Jorma J. Palvimo[5]

Androgen receptor (AR) is regulated by SUMOylation at its transactivation domain. In vitro, the SUMOylation is linked to transcriptional repression and/or target gene-selective regulation. Here, we generated a mouse model (ArKI) in which the conserved SUMO acceptor lysines of AR are permanently abolished ($Ar^{K381R, K500R}$). ArKI males develop normally, without apparent defects in their systemic androgen action in reproductive tissues. However, the ArKI males are infertile. Their spermatogenesis appears unaffected, but their epididymal sperm maturation is defective, shown by severely compromised motility and fertilization capacity of the sperm. Fittingly, their epididymal AR chromatin-binding and gene expression associated with sperm maturation and function are misregulated. AR is SUMOylated in the wild-type epididymis but not in the testis, which could explain the tissue-specific response to the lack of AR SUMOylation. Our studies thus indicate that epididymal AR SUMOylation is essential for the post-testicular sperm maturation and normal reproductive capability of male mice.

[1] Institute of Biomedicine, Research Centre for Integrative Physiology and Pharmacology, University of Turku, FI-20520 Turku, Finland. [2] Turku Center for Disease Modeling (TCDM), Institute of Biomedicine, University of Turku, FI-20520 Turku, Finland. [3] Department of Environmental and Biological Sciences, University of Eastern Finland, FI-80101 Joensuu, Finland. [4] Turku Centre for Biotechnology, University of Turku and Åbo Akademi University, FI-20014 Turku, Finland. [5] Institute of Biomedicine, University of Eastern Finland, FI-70211 Kuopio, Finland. [6] Institute of Medicine, The Sahlgrenska Academy, Gothenburg University, 41345 Gothenburg, Sweden. These authors contributed equally: Petra Sipilä, Jorma J. Palvimo. Correspondence and requests for materials should be addressed to J.J.P. (email: jorma.palvimo@uef.fi)

SUMOylation, covalent conjugation of small ubiquitin-related modifiers (SUMOs), is a conserved posttranslational modification. It has emerged as a widespread and important regulatory mechanism in cell physiology, especially in DNA damage, stress responses, nuclear signaling and in transcription[1–4]. SUMO modifications have also been associated with many disease conditions, ranging from host antibacterial response, cardiovascular and neurodegenerative diseases to cancer[4–8]. SUMOylation functions by providing docking sites for interacting proteins, which leads to altered cellular localization, activity and/or stability of the target proteins. Mammalian cells possess three SUMO isoforms, SUMO-1, -2 and -3, that can be conjugated to proteins. They are ca. 100-amino acid long proteins that possess the ubiquitin fold. SUMO-1 is ~50% identical with SUMO-2 and -3 that are nearly identical (97%), and therefore often collectively referred to as SUMO-2/3[1,4]. The SUMOylation process is similar to ubiquitylation, but it requires a distinct enzymatic cascade. SUMOs activated by SAE1/2 heterodimer are conjugated to target lysines by UBC9. SUMO ligases, such as PIAS proteins, can assist SUMOylation by guiding the target lysine to the active site of the UBC9. Most of the target proteins show a low stoichiometry of SUMOylation, which is thought to reflect dynamic SUMOylation-deSUMOylation cycles of the target proteins. The deSUMOylation is catalyzed by a family of SUMO-specific proteases, SENPs[9]. Knockouts of Ubc9 and Sumo-2 are embryonic lethal in mouse[10,11]. Lack of severe phenotypes in Sumo-1 or Sumo-3 knockout mice is likely to be due to compensation by Sumo-2[11,12].

Androgen receptor (AR) was among the first transcription factors shown to be SUMOylated[13]. The AR plays, among other actions, a crucial role in the development of male phenotype and reproductive functions[14]. It is principally activated by binding of ligand, but the function of androgen-activated receptor is further regulated by posttranslational modifications, including SUMOylation[13,15]. Based on in vitro studies with reporter genes, the SUMOylation was initially postulated to simply repress the transcriptional activity of the receptor[13,16]. Recently, however, genome-wide gene analyses of prostate cancer cells expressing SUMOylation site-deficient AR showed that the SUMOylation regulates the receptor's function in a target gene-specific manner. Particularly the androgen target genes associated with the control of cell cycle, cell proliferation, and cell death were shown to be sensitive for SUMOylation-mediated regulation, which was reflected in apoptosis sensitivity and proliferation of the cancer cells expressing the mutant receptor[17]. SUMOylation similarly modulates the activity of progesterone receptor and that of glucocorticoid receptor[18,19], suggesting that SUMOylation widely regulates the action of the transcription factors activated by 3-keto steroids. However, the biological consequences of SUMOylation (or posttranslational modifications in general) of these nuclear receptors in vivo have remained to be determined.

In this work, we have generated a knockin mouse model (ArKI mice) in which the conserved SUMO acceptor lysines in the N-terminal transactivation domain of the AR are permanently abolished by converting them to arginines ($Ar^{K381R,\ K500R}$). Interestingly, despite apparent lack of system-wide effects on androgen signaling in several peripheral tissues, ArKI males are infertile. Detailed analysis of the mutant mice reveals that the conserved SUMOylation sites of AR that show SUMO-2/3 modification in wild-type (WT) epididymis are essential for the proper gene expression in the epididymis and male fertility.

## Results

### Generation of AR SUMOylation-deficient mice
Human AR has been shown to be SUMOylated at K386 and K520[13]. The lysine residues are embedded in perfect SUMOylation consensus sequences that are both fully conserved in mammalian AR sequences, corresponding to K381 and K500 in mouse AR. To abolish AR SUMOylation in mouse, the lysines were mutated to arginines to generate $Ar^{K381R,K500R}$ knockin (ArKI) mice (Fig. 1a). The mouse Ar gene is located in chromosome X, and therefore, the mutation of only one allele is enough to replace the WT AR with the mutated form in males. The mice were genotyped by PCR (Fig. 1b). RT-qPCR and immunoblotting showed that the mutant AR is expressed at a level comparable with that of WT AR in the testis and epididymis (Fig. 1c, d). Moreover, immunofluorescent staining revealed that the subcellular distribution of the mutant does not differ from that of the WT AR in the epididymis (Fig. 1e) or the testis (Supplementary Figure. 1a).

### AR SUMOylation-deficient mice develop normally
ArKI male mice were born phenotypically normal, and the body weights of ArKI and WT littermates over a 10-month period were indistinguishable (Fig. 2a). Testes had descended normally into the scrotum, and there was no difference in ano-genital distance between ArKI and WT mice at 30 and 60 d of age (Fig. 2b). The testes and the accessory reproductive organs (epididymis, seminal vesicle and prostate) were macroscopically normal (Fig. 2c). Furthermore, tissue weights of the peripheral androgen-dependent tissues in adults were comparable between WT and ArKI mice (Fig. 2d). In line with this, serum gonadotropin and steroid hormone levels were similar between WT and ArKI male mice (Supplementary Table 1). Furthermore, there were no statistically significant differences in intraepididymal and intratesticular steroid levels in ArKI males compared to WT (Supplementary Table 1).

### AR SUMOylation is required for the normal sperm functions
Androgens and AR are imperative for both development and maintenance of male reproductive functions. We, therefore, scrutinized whether the fertility of male mice is affected by the mutation of AR SUMOylation sites. Interestingly, breeding of ArKI males with WT females yielded no offspring. However, we were not able to detect any vaginal plugs, suggesting that ArKI males did not copulate with females, or their semen was altered in a fashion that it did not form a copulatory plug. Of note, in some ArKI males, the seminal vesicle composition was indeed observed to be unusually solid, supporting the latter explanation.

Next we performed functional analyses of ArKI sperm in order to characterize the fertility defect in more detail. Interestingly, compared to WT sperm, a significantly lower portion of ArKI sperm was moving progressively forward ($33.2 \pm 7\%$ vs. $71.3 \pm 3\%$, $p \leq 0.001$, two-tailed Student's t test, mean ± SEM, $n = 6$ each genotype) and notably, the majority of ArKI sperm was totally immotile ($55.7 \pm 8\%$ vs. $18.3 \pm 3\%$, $p \leq 0.001$, two-tailed Student's t test) (Fig. 2e). Furthermore, in in vitro fertilization test, sperm from ArKI mice showed a significantly reduced ability to fertilize oocytes (percentage of fertilized oocytes: WT $59.7 \pm 2.67\%$ vs. ArKI $8.8 \pm 0.74\%$, $p \leq 0.001$, two-tailed Student's t test, mean ± SEM, $n = 6$ each genotype) (Fig. 2f). There was no difference in the number of acrosome-reacted sperm between ArKI and WT mice when utilizing a calcium ionophore or ZP proteins as inducers (Fig. 2g), indicating that the lowered in vitro fertilization capacity is not due to the defective acrosome reaction. These results show that the AR SUMOylation is critical for normal sperm maturation and thus, male fertility.

### Epididymal histology is affected by AR SUMOylation
To study spermatogenesis and epididymal sperm maturation, both androgen-dependent processes, in more detail, we analyzed the

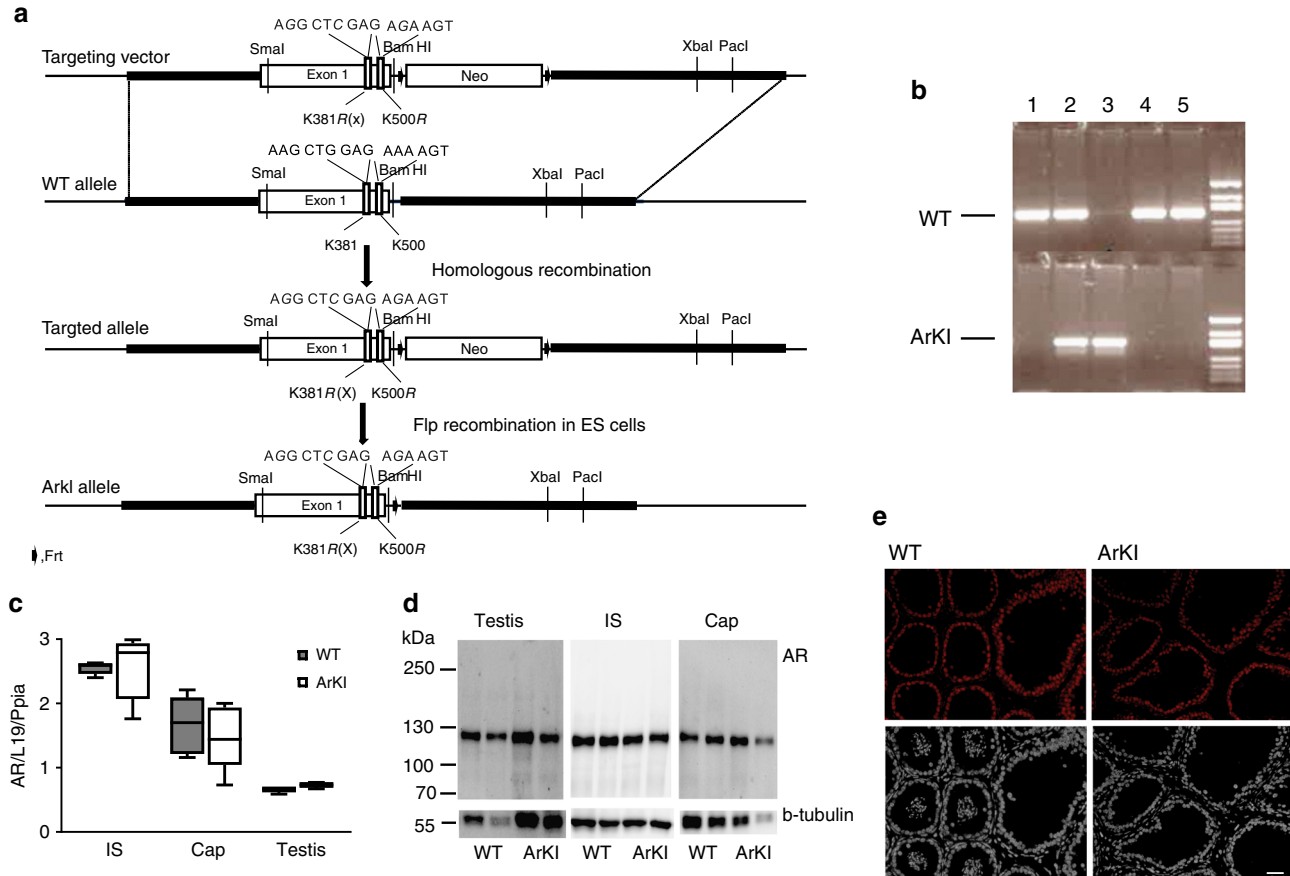

**Fig. 1** ArKI targeting construct and expression of mutated AR. **a** Schematic representation of ArKI targeting construct. Two major SUMOylation sites, lysine in K381, K550 were mutated to arginines (K381R, K550R). X, generated *Xho*I site. **b** PCR used to detect WT and KI allele (lines 1–2 female; lines 3–5 male). **c** RT-qPCR analysis of *Ar* mRNA levels between ArKI and WT initial segment (IS) and caput (Cap) epididymidis and testis (*n* = 5 WT and 6 ArKI mice). Boxplot center lines indicate the median and the boxes extend to lower and upper quartiles with whiskers depicting maximum and minimum. **d** Western blotting of AR in ArKI and WT epididymis and testis. Immunoblotting of β-tubulin levels was used to control protein loading. **e** Immunofluorescent localization of AR in the IS/Cap epididymidis. Upper panels (red), AR; lower panels (white), DAPI. Scale bar 50 μm. AR androgen receptor, WT: wild type, KI: knockin, DAPI: 4′.6-diamidino-2-phenylindole

histology of the testes and the epididymides of ArKI mice. In the adult testis, all stages of the seminiferous epithelial cycle were identified, and no visible morphological defects were observed in differentiating male germ cells (Supplementary Figure 1b, c). However, seminiferous tubule lumens appeared to be expanded in about half of the ArKI males, which may indicate defects in Sertoli cell secretion or sperm flow though the epididymis. Acrosomal staining and immunostaining of ESPIN that visualizes the cell–cell junctions between Sertoli cells and those between Sertoli cells and elongating spermatids were used to confirm the WT-like organization of the seminiferous epithelium in ArKI mice (Supplementary Figure 2a). No defects were observed in the nuclear elongation or condensation of spermatids (Supplementary Figure 2b), and the morphology of cauda epididymal spermatozoa appeared normal (Supplementary Figure 2c). Furthermore, the first wave of spermatogenesis appeared to progress normally, as demonstrated by the unaffected testis histology in juvenile 2-, 3-, 5- and 7-week-old ArKI mice compared to WT mice (Supplementary Figure 3).

Histological analyses from juvenile 2-, 3-, 5- and 7-week-old mice revealed that the ArKI epididymides were indistinguishable from WT epididymides at the ages of 2, 3 and 5 weeks (Supplementary Figure 4a). At seven weeks of age 67–100% of animals had few first vacuoles, mild epithelial hyperplasia and folding, and enlarged afferent ducts. Epididymides of 3-month-

old ArKI mice displayed mild histological changes with varying frequency (Fig. 3f). In contrast, 100% of the mutant animals (0% of the WT mice) displayed vacuoles specifically at the caput segments 4 and 5 (segment numbers according to Abou-Haïla and Fain-Maurel[20]) (Fig. 3a, f). Other frequent features of the ArKI epididymides included enlarged efferent ducts, stromal hyperplasia, dilated epithelial tubules and inflammation lymphocytes. The phenotypic changes became more pronounced with the age in 6 to 7- and 10-month-old mice (Fig. 3f–h) and included dilated efferent ducts (Fig. 3b), epithelial hyperplasia with papillary features (Fig. 3c), nuclear atypia (Fig. 3a) and chronic inflammation characterized by lymphocyte infiltration and fibrosis (Fig. 3d). Some individuals showed no sperm in epididymal lumen (Fig. 3f–h). Along with observed expansion of seminiferous tubule lumens, these aberrancies indicate obstructions in the efferent ducts or the proximal epididymal ducts.

To study further how the lack of AR SUMOylation affects epididymal cell differentiation, we analyzed the presence of epididymal basal and smooth muscle cells. Antibody against KRT5 was used to stain basal cells, and antibody against SMA was used to visualize the smooth muscle cell layer surrounding the epithelial tubes. There were no apparent changes in the structure of the muscle cell layer (Fig. 3e). However, the basal cell layer located at the basolateral side of the epithelial cell layer was

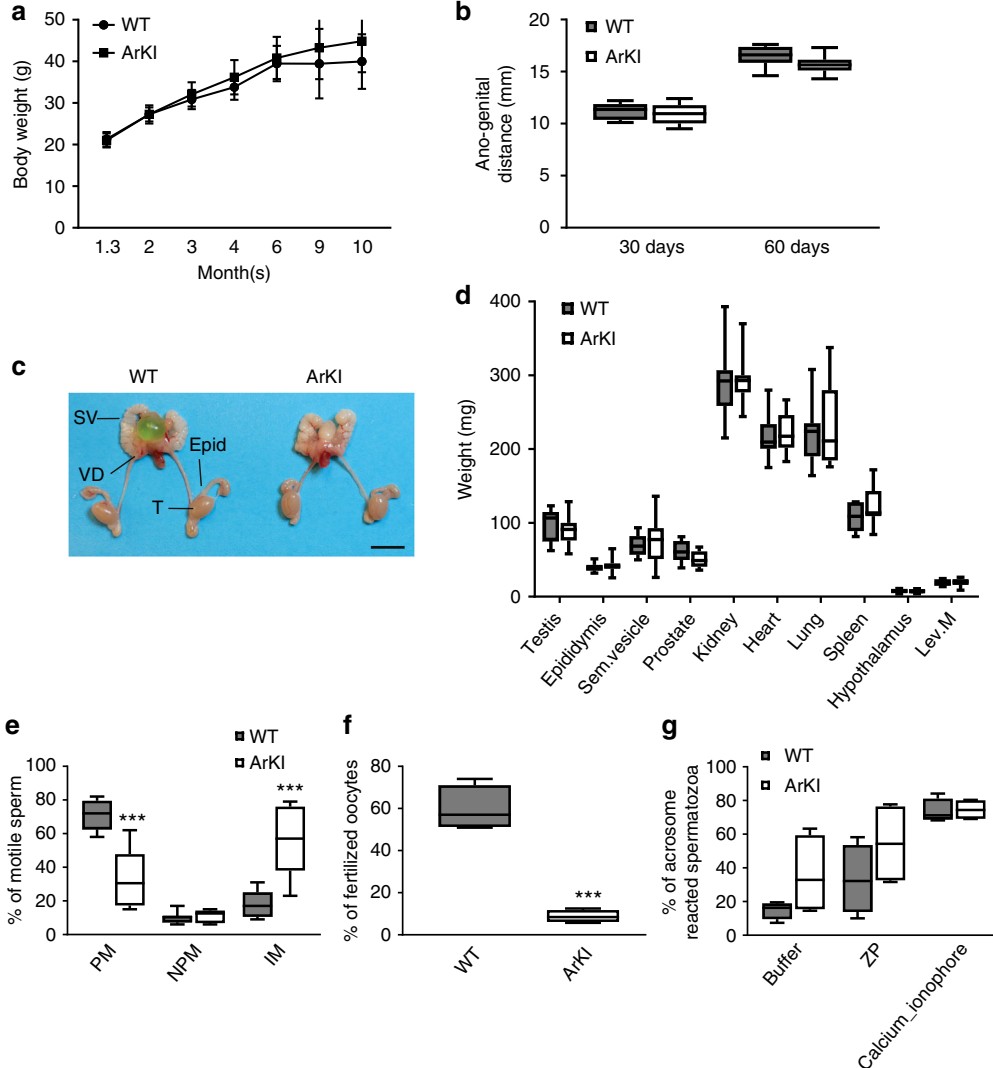

**Fig. 2** Effects of AR SUMOylation on peripheral target tissues, sperm motility and fertilizing capacity. **a** Postnatal growth of ArKI mice and WT mice. Body weights of ArKI mice and WT mice were recorded over a 10-month period. Data presented as mean ± SD of 12 WT and 16 ArKI mice. **b** Ano-genital distance of ArKI and WT mice at the age of 30 and 60 days ($n = 12$ WT and 11 ArKI males). **c** Macroscopic phenotype of testis and accessory sex organs of ArKI male and WT littermate. T: testis, Epid: epididymis, VD: vas deferens, SV: seminal vesicle. Scale bar 1 cm. **d** Organ weights of ArKI mice and WT mice at the age of 3 months ($n = 10$ for each genotype). **e** Sperm motility shown as percentage of sperm with progressive (PM) or nonprogressive motility (NPM) and percentage of immotile sperm (IM) ($n = 6$ males for each genotype). ***$p < 0.001$. **f** In vitro fertilization data represented as percentage of oocytes fertilized by sperm from ArKI mice and WT control mice ($n = 6$ mice for each genotype). ***$p < 0.001$. **g** Analysis of the acrosome reaction presented as percentage of acrosome-reacted spermatozoa after incubation with buffer alone, mouse zona pellucida (ZP) or calcium ionophore ($n = 4$ mice for each genotype). Boxplot center lines indicate the median and the boxes extend to lower and upper quartiles with whiskers depicting maximum and minimum. $p$ values were determined by two-tailed Student's $t$ test. AR: androgen receptor, WT: wild type

scarcer and lacked long and narrow cytoplasmic projections typically reaching towards the lumen (Fig. 3e). Taken together, the defective AR SUMOylation induced morphological changes in the epididymis, therefore compromising reproductive functions.

**Lack of AR SUMOylation alters epididymal gene expression**. To understand how the lack of AR SUMOylation affects gene expression in the epididymis and the testis, we compared the transcriptome profiles of ArKI and WT mice using RNA-seq. Interestingly, pronounced changes in the gene expression were observed in the ArKI epididymis: 235 genes were significantly downregulated in the initial segment (IS) and 511 genes in the caput, whereas 376 genes were significantly upregulated in the IS

and 236 genes in the caput. Notably, the unsupervised hierarchical clustering of known androgen-regulated genes in the epididymis showed that a subset of androgen-regulated genes in the IS and caput epididymidis was indeed differentially expressed between ArKI and control mice (Fig. 4a). From a total of 778 androgen-regulated genes, 117 genes were differentially expressed (false discovery rate (FDR) < 0.05 and absolute fold-change ≥ 2) in ArKI epididymides (Supplementary Data 1). For example, the expression of *Rhox5*, an androgen-regulated homeodomain transcription factor and several members of the Spink family of serine peptidase inhibitors were significantly altered in ArKI epididymides based on RNA-seq data. RT-qPCR results confirmed the attenuated expression of *Rhox5*, *Spink2, -5* and *-11* in ArKI (IS and/or caput) epididymides (Fig. 4b, c). Furthermore, *Spink4, -12* and *-13* that showed a tendency of being

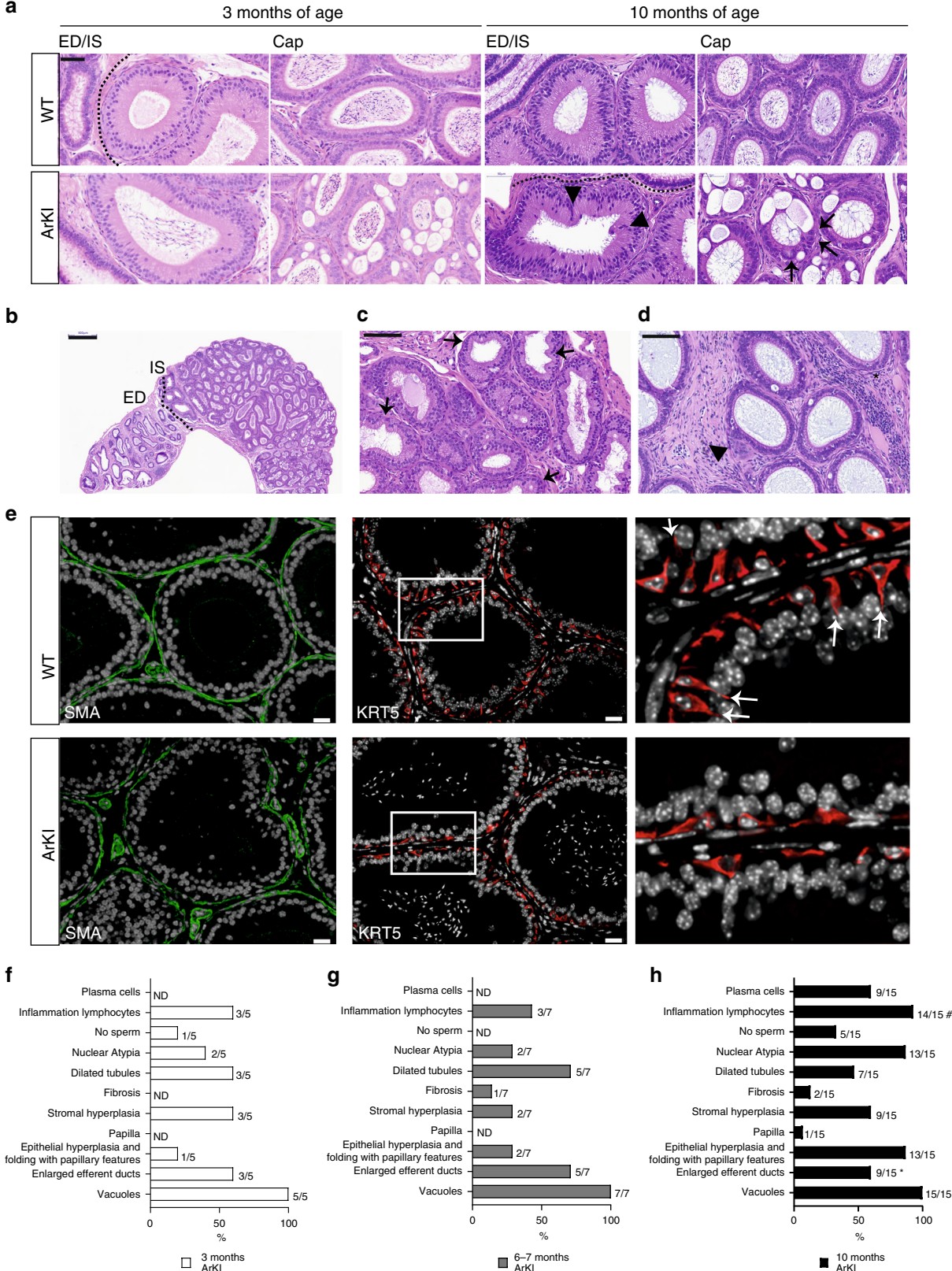

downregulated in RNA-seq were significantly changed based on RT-qPCR data (Fig. 4b, c). *Spink* genes were also downregulated at earlier time points, 3 and 5 weeks of age (Supplementary Figure 4b), when histological appearance of epididymal epithelium was still normal in ArKI males. Interestingly, the expression

of *Rhox5* and *Spink2* that have known functions in the testis were not significantly changed in ArKI testes (Fig. 4d), suggesting differential regulation of these genes in the testis and epididymis. Overall, the abolishment of AR SUMOylation had only minor effects on the testis gene expression profile (Supplementary

**Fig. 3** Epididymal histology is disturbed in ArKI males. **a** Hematoxylin and eosin staining of WT and ArKI mouse efferent ducts (ED), initial segment (IS) and caput (Cap) epididymides at the age of 3 and 10 months. At 10-month-old ArKI IS the epithelial hyperplasia with papillary features is marked with arrow heads and ArKI Cap with nuclear atypia are marked with arrows. Scale bar 50 µm. **b** Enlarged efferent ducts (ED). Initial segment (IS). Scale bar 500 µm. **c** Epithelial hyperplasia with papillary features (arrows). Scale bar 100 µm. **d** Chronic inflammation characterized by lymphocyte infiltration (star) and fibrosis (arrow head). Scale bar 100 µm. **e** Immunofluorescent staining of the epididymal basal and smooth muscle cells. Staining of initial segment (IS) of WT and ArKI mouse with smooth muscle actin (SMA, green) for smooth muscle cells and keratin 5 (KRT5, red) for basal cells. DAPI was used for staining nuclei (white). Scale bars 20 µm. Right panel shows higher magnification of the area marked with box in KRT5 stained middle panel. White arrows indicate basal cell projections that extend towards the lumen. **f–h** Histological findings in ArKI epididymis at different ages. The numbers next to the bars refer to phenotypes observed/total number of samples at that age group. ND: not detected. *Observed also in 11% (1/9) of 10-month-old WT epididymides. #Observed also in 22% (2/9) of 10-month-old WT epididymides. WT: wild type, DAPI: 4′.6-diamidino-2-phenylindole

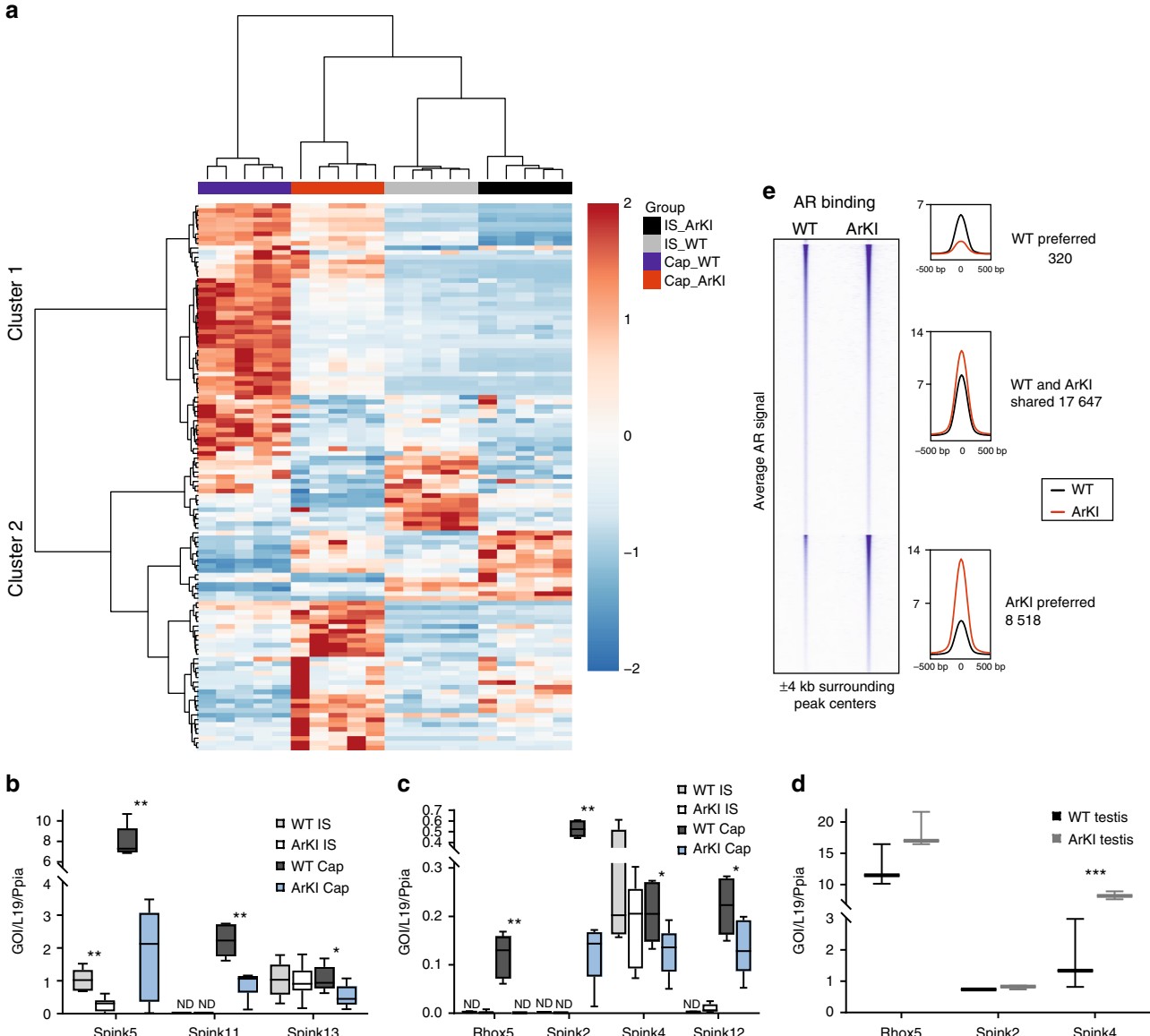

**Fig. 4** Lack of AR SUMOylation alters epididymal gene expression and AR chromatin binding. **a** Heat map of known androgen-regulated genes clustered by unsupervised hierarchical clustering from RNA-seq data. **b** Relative expression of *Spink5*, *Spink11* and *Spink13* mRNA in 3-month-old WT and ArKI IS and Cap epididymides by RT-qPCR normalized to *L19* and *Ppia* expression. *$p < 0.05$, **$p < 0.01$. **c** Relative expression of *Spink2*, *Spink4*, *Spink12* and *Rhox5* mRNA in WT and ArKI IS and Cap epididymides by RT-qPCR normalized to *L19* and *Ppia* expression. *$p < 0.05$, **$p < 0.01$. **d** Relative expression of *Spink2*, *Spink4* and *Rhox5* mRNA in WT and ArKI testis by RT-qPCR. ***$p < 0.001$. In panels **b–d** $n = 5$ WT and 6 ArKI mice. Boxplot center lines indicate the median and the boxes extend to lower and upper quartiles with whiskers depicting maximum and minimum. $p$ values were determined by two-tailed Student's $t$ test or by Mann−Whitney test. ND not detected. **e** Heat map showing AR ChIP-seq tag densities in WT and ArKI epididymis at WT AR-preferred, WT and ArKI-shared and ArKI-preferred AR chromatin-binding sites (ARBs) in a window ±4 kb (left panel). Comparison of the WT and the ArKI average tag counts ±500 bp from the centers of the ARBs in the three categories (right panel). AR: androgen receptor, WT: wild type

**Table 1 Top 20 enriched GO terms among differentially expressed caput mRNAs in ArKI epididymis**

| GO.ID | Term | p value |
|-------|------|---------|
| GO:0019953 | Sexual reproduction | 3.6e-20 |
| GO:0007283 | Spermatogenesis | 1.3e-19 |
| GO:0048232 | Male gamete generation | 1.5e-19 |
| GO:0044703 | Multiorganism reproductive process | 5.6e-18 |
| GO:0007276 | Gamete generation | 6.6e-17 |
| GO:0032504 | Multicellular organism reproduction | 1.3e-15 |
| GO:0048609 | Multicellular organismal reproductive process | 9.3e-15 |
| GO:0022414 | Reproductive process | 4.00E-14 |
| GO:0000003 | Reproduction | 4.2e-14 |
| GO:0044702 | Single organism reproductive process | 1.00E-13 |
| GO:0030317 | Flagellated sperm motility | 7.9e-12 |
| GO:0097722 | Sperm motility | 7.9e-12 |
| GO:0009566 | Fertilization | 1.1e-07 |
| GO:0051704 | Multiorganism process | 1.3e-07 |
| GO:0022412 | Cellular process involved in reproduction... | 2.7e-07 |
| GO:0007286 | Spermatid development | 3.00E-07 |
| GO:0048515 | Spermatid differentiation | 6.6e-07 |
| GO:0007281 | Germ cell development | 1.4e-06 |
| GO:0007338 | Single fertilization | 1.8e-05 |
| GO:0086010 | Membrane depolarization during action potential | 0.00012 |

GO: gene ontology

Data 1). The few differentially expressed genes in the testis included *Spink4* whose upregulation was also confirmed by RT-qPCR (Fig. 4d). In line with the phenotypic changes in the mutant mice, gene ontology (GO) term analysis of all the up- and downregulated genes in the ArKI epididymides revealed a significant enrichment of GO terms related to sperm function and fertilization process in the caput region (Table 1). The evident changes in the gene expression suggest that the SUMOylation has an important role in the regulation of AR-dependent transcription in the epididymis.

**SUMOylation regulates AR binding to epididymal chromatin**. To complement the RNA-seq analysis in the epididymis, we investigated whether the SUMOylation affects AR chromatin occupancy by using chromatin immunoprecipitation coupled to deep sequencing (ChIP-seq). Interestingly, the ChIP-seq analyses revealed more intense chromatin occupancy of the receptor in ArKI epididymis (Fig. 4e), and the number of chromatin-binding sites (ARBs) in the ArKI epididymis (26,165) was nearly 1.5-fold compared to that of WT epididymis (17,967). Although most of the ARBs (17,647, termed WT- and ArKI-shared ARBs) overlapped between the ArKI and WT epididymis, there were 8518 AR SUMOylation mutant-preferred sites, showing at least two-fold enhanced occupancy of the SUMOylation mutant compared to that of the WT AR, whereas conversely, the number of WT AR-preferred ARBs (320) was minuscule (Fig. 4e). Many genes that were upregulated in ArKI epididymis showed, in addition to AR SUMOylation mutant-preferred ARBs, a stronger binding of the mutant AR to the shared ARBs (Supplementary Figure 5a, b) or even an ARB not detectable with the WT AR (Supplementary Figure 5c). A similar situation was also seen with some genes downregulated in ArKI epididymis, such as *Spink11* (Supplementary Figure 5d), suggesting that enhanced chromatin interaction of SUMOylation mutant AR does not simply explain the altered expression of AR target genes in the ArKI epididymis. Integrating the AR ChIP-seq data sets with the corresponding RNA-seq data sets confirmed significantly enhanced binding of

the SUMOylation mutant AR to the differentially regulated genes, with the difference being more significant with upregulated genes ($p < 7.238$e-15, Fisher's test) than downregulated genes ($p < 0.0004375$, Fisher's test). De novo motif analyses of the all ARBs to identify transcription factor-binding motifs enriched within them revealed, as expected, ARE (androgen response element) consensus motif as the most enriched motif among both WT- and ArKI-shared ARBs and ArKI-preferred ARBs (over 80%, Supplementary Table 2). However, there were differences between the ARB categories in the motif enrichment, as NANOG and ELF5 motifs listed in the top ten enriched motifs among the ArKI-preferred ARBs and EGR1 motif among the top ten shared ARBs (Supplementary Table 2). Thus, our ChIP-seq analyses indicate that the SUMOylation regulates AR's interactions with the epididymal chromatin in vivo. The perturbed chromatin binding of the mutant is likely to reflect altered interactions of the AR mutant with chromatin proteins and other transcription factors and thereby contribute to the dysregulated gene expression in the ArKI epididymis.

**AR is differentially SUMOylated in epididymis and testis**. In order to understand why the abolishment of the SUMOylation sites of AR affected the function of the epididymis without obvious consequences in another highly androgen-dependent target tissue, the testis, we studied the levels of AR SUMOylation in these tissues in adult animals. Immunoprecipitation of AR followed by immunoblotting with antibodies against SUMO-1 or SUMO-2/3 revealed that the modification of the AR by SUMO-2/3 (but not by SUMO-1) was clearly detectable in the WT epididymis but not in the WT testis. As expected, no AR SUMOylation was detected in the ArKI epididymis due to the mutated SUMO acceptor lysines (Fig. 5a). Analysis of younger (2-, 3-, 5-, 7- and 10-week old) mice demonstrated that regardless of hormonal status (prepuberty, puberty, postpuberty) or differentiation phase of epididymal epithelium, AR is SUMOylated in the epididymis but not in the testis (Fig. 5b). While the reason for the more pronounced SUMOylation of AR in the epididymis compared to the testis remains to be characterized, one explanation may be the clearly higher expression of SUMO/sentrin specific peptidase 1 and -2 (*Senp1*, *Senp2*) and desumoylating isopeptidase 1 (*Desi1*) in the testis (Supplementary Table 3). As SENP1 and the SENP2 are efficient in reversing AR SUMOylation[16], it is possible that their high expression renders the AR SUMOylation relatively unstable in the testis.

**Discussion**

In order to study the biological role of AR SUMOylation in vivo, we generated a knockin mouse model in which the conserved SUMO attachment lysines of AR were permanently abolished by converting them to arginines. Even though we cannot fully exclude the possibility that the mutated lysines would affect other posttranslational modifications, no other modifications or roles for these conserved lysines have been suggested. Recently, the SUMOylation acceptor lysine residues were mutated in the context of polyglutamine (polyQ) track-expanded (113Q) AR, a mouse model of spinobulbar muscular atrophy (SBMA)[21]. Disrupting SUMOylation in the context of AR113Q rescued exercise endurance and type I muscle fiber atrophy, prolonging also survival of the SBMA mice[21]. Compared to WT mice, both AR113Q mice and AR113Q SUMOylation-deficient mice showed testicular and seminal vesicle atrophy and reduced sperm counts. However, mutation of the SUMOylation sites in normal polyQ track-containing AR in this study did not have apparent effects on the development and growth of ArKI male mice. Interestingly, despite the normal appearance of many peripheral androgen target tissues, the AR SUMOylation-deficient

 

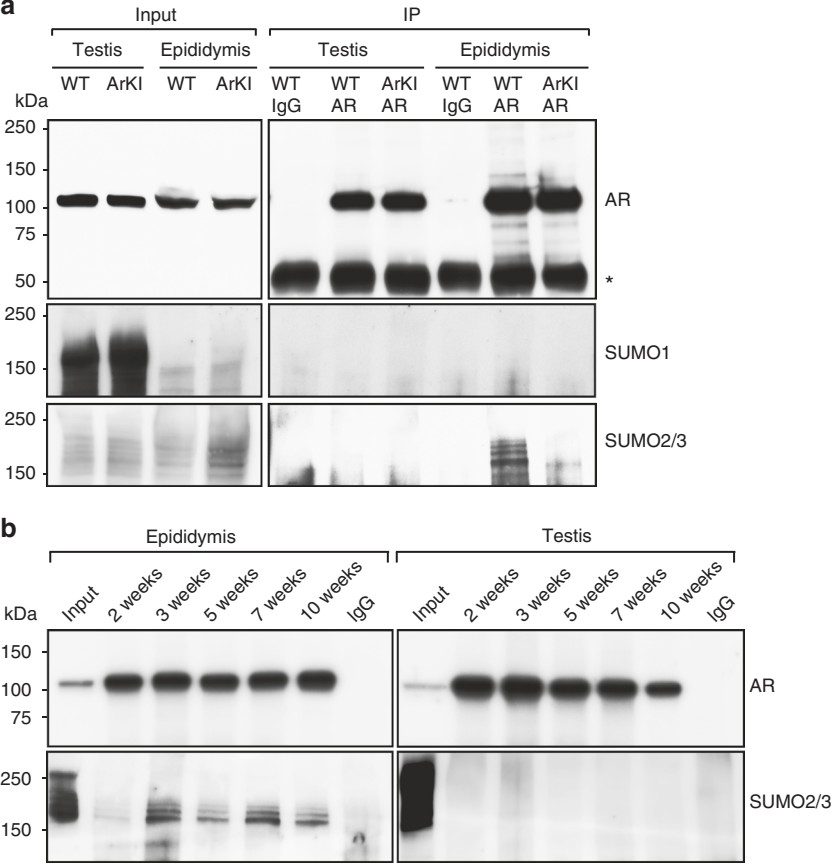

**Fig. 5** AR SUMO-2/3 modification is detectable in epididymis but not in testis. **a** AR, SUMO-1 and SUMO-2/3 were immunoblotted from the input lysate samples, anti-AR-immunoprecipitates and IgG-immunoprecipitates (controls) of 3-month-old ArKI and WT mice. An asterisk indicates the IgG heavy chain band. The experiment was repeated three times with comparable results. **b** AR SUMO-2/3 modification is also detectable between 2 and 10 weeks of age in the epididymis, but not in the testis

male mice were infertile. They showed morphological changes in the epididymis and their sperm motility and ability to fertilize oocytes in vitro were severely impaired, indicating that the AR SUMOylation sites are critical for normal sperm maturation. The reduced motility of the spermatozoa of ArKI mice resemble some of the phenotypic characteristics of specificity-affecting AR knockin (SPARKI) mice, in which the AR DNA-binding domain was mutated in a fashion that it lost binding to direct repeats but not to inverted repeats[22]. Despite the SPARKI mice displayed similar defects in their sperm motility as the AR SUMOylation-deficient mice, the underlying molecular mechanisms for these defects appear to be different in these mice, as the mutations resulted in distinct changes in the gene expression in the epididymides[22,23]. From over 200 androgen-regulated genes altered in SPARKI epididymis, only ten were shared with ArKI mice (Supplementary Table 4).

There are only two previous studies addressing the effect of SUMOylation of an endogenous substrate in vivo. Eliminating the SUMOylation of steroidogenic factor 1 (SF-1) resulted in ectopic hedgehog signaling and disruption of endocrine development in mice[24]. Unlike AR, SF-1 is not activated by a switchlike ligand. Thus SUMOylation of two conserved lysines residing at the SF-1's hinge region, adjacent to the DNA-binding domain and ligand-binding domain, seem to be particularly important for the proper function of this orphan nuclear receptor. The mechanism underlying the disrupted endocrine development of SUMOylation-deficient SF-1 mice was associated with inappropriate activation

of SF-1 target genes during the development. In the case of another orphan nuclear receptor, liver receptor homolog-1 (LRH-1), a mutation that abolishes SUMOylation at its hinge region in atherosclerosis-prone mice resulted in reduced number of aortic plaques compared to the control littermates when exposed to a high-cholesterol diet[25]. The mechanism underlying this atheroprotection involved a SUMOylation-dependent interaction of the LRH-1 with the corepressor prospero homeobox protein 1 (PROX1).

Our RNA-sequencing-based expression profiling of ArKI mice revealed that the abnormal fertility and sperm function is associated with significantly altered expression of a subset of AR target genes in the epididymis, whereas there were minor changes in the testicular gene expression. Based on the analyses of AR SUMOylation, the lack of pronounced testis phenotype in ArKI mice is likely to derive from tissue-specific differences in the modification of AR by SUMO-2/3. A number of genes with altered expression in ArKI epididymis associated with sperm function and fertilization processes. This group included *Rhox5* (previously known as *Pem*) encoding an androgen-regulated homeobox transcription factor expressed in testicular Sertoli cells and caput epididymidis[26]. Interestingly, Rhox5 knockout mice were subfertile due to increased germ cell apoptosis with subsequent reduction in sperm count and further increase in the percentage of immotile caudal spermatozoa[27]. Sperm motility defect in Rhox5 knockout mice was suggested to be of epididymal origin[26]. These findings are in line with our results. In ArKI

males, the testicular *Rhox5* expression was not changed, whereas epididymal *Rhox5* expression was significantly downregulated, correlating well with normal testicular phenotype but disturbed epididymal functions and defect in sperm motility. Also the aberrant expression of the Spink family genes in the ArKI mice is likely to contribute to the infertility and defective sperm motility. Several SPINKs are expressed in the male reproductive tract: *Spink2* in testis, *Spink3* in seminal vesicle and *Spinks* -8, -10, -11, -12 and -13 in epididymis, and all the epididymal *Spinks* are androgen-dependent[28,29]. From the *Spink* genes present in reproductive tract, the *Spink2* haploinsufficiency has been shown to result in a reduced sperm motility due to morphological defects[30]. However, epididymal sperm maturation involves the processing of the sperm membrane by several proteases, and thus, the many epididymal *Spinks* may have an important role as protease inhibitors to ensure correct spatio-temporal control of sperm membrane processing. We therefore suggest that in addition to the *Rhox5*, the concurrent downregulation of six epididymal SPINKs contributes to the observed motility and fertilizing ability defects in ArKI sperm. However, despite no apparent testicular phenotype in the ArKI mice, we cannot formally exclude the possibility that their altered testicular gene expression also in part contributes to their compromised sperm function.

Our genome-wide AR chromatin-binding analyses of ArKI vs. WT mice also revealed that the SUMOylation restricts the AR's binding to chromatin in the epididymis in vivo. Similarly, the modification influences the interaction of the receptor with the chromatin and regulation of its target genes in prostate cancer cells[17]. The enhanced interaction of the SUMOylation mutant AR with the epididymal chromatin does not however offer a straightforward explanation for the altered AR target gene expression in the ArKI mice. Nevertheless, the perturbed chromatin binding is likely to reflect changed interactions of the AR mutant with chromatin proteins and other transcription factors and thereby contribute to the dysregulated androgen regulation of the epididymis of the ArKI mice.

Taken together, our in vivo analyses reveal that the SUMOylation status of AR regulates the function of the transcription factor in a tissue-specific fashion, playing an important role in controlling the effects of androgens on proper epididymal functions and maturation of spermatozoa. Interestingly, several human AR mutations associated with oligospermia and androgen insensitivity syndrome map to a conserved amino acid just downstream of the primary SUMOylation motif in AR and they attenuate the AR SUMOylation, implying clinical relevance of our findings[31].

## Methods

**Generation of the ArKI mouse line**. K381 and K500 (corresponding to human AR K386 and K520)[13] were converted to R381 and R520 in the endogenous *Ar* mouse locus (Fig. 1a). BAC clone (RP23-243G4) containing mouse *Ar* gene was obtained from BACPAC Resources Center (Children's Hospital Oakland Research Institute, Oakland (CHORI), California, USA), and a 8540 bp fragment of the *Ar* gene containing exon 1 was subcloned into the minimal vector (pACYC177, New England Biolabs, MA, USA) by Red/ET recombination according to the manufacturer's instructions (Gene Bridges GmbH, Germany). Site-directed mutagenesis was then performed to create the Ar^K381R and Ar^K500R mutations and to introduce an *XhoI* site (Fig. 1a). The Neo cassette flanked with two Frt sites was inserted into the intron 1 at site around 200 bp after exon1 by Red/ET recombination. The full targeting construct was verified by sequencing. All primer sequences used in cloning are in Supplementary Table 5.

Hybrid mouse embryonic stem (ES) cells (G4, 129S6B6F1) were electroporated with the linearized targeting construct. The targeted ES cell clones were screened by long-range PCR and further confirmed by sequencing. FRT-flanked Neo cassette was removed from correctly targeted ES cells using FLP/FRT site-specific recombination in vitro. Subsequent ES cell colonies were screened by PCR with several different primer pairs and further confirmed by sequencing. All primer sequences used in screening are in Supplementary Table 5.

The targeted clones with *Neo* deletion were injected in C57BL/6N blastocysts, and resulting chimeric males were subsequently bred to C57BL/6N females to test

germline transmission. F1 heterozygous females were bred with wild-type (WT) C57BL/6N male in order to get male mice carrying mutated allele on X chromosome (F2). All mice were handled in accordance with the institutional animal care policies of the Central Animal Laboratory, University of Turku (Turku, Finland). The mice were specific pathogen-free, fed with pelleted soy-free natural-ingredient feed (RM3 (E), Special Diets Services, Essex, UK) and tap water ad libitum, and housed in individually ventilated cages under controlled conditions of light, temperature, and humidity. Animals were analyzed at following age points: at 2 weeks (prior puberty: the first wave of spermatogenesis in the testis has reached pachytene spermatocyte phase, Sertoli cells are still immature, epididymal epithelium and segments are undifferentiated), at 3 weeks (prior puberty: early haploid cells have appeared in the testis, epididymal epithelium and segments start to differentiate), at 5 weeks (puberty: the first wave of spermatogenesis is completed, epididymal differentiation continues), at 7 weeks (full spermatogenesis ongoing in the testis, epididymal epithelium and segments are fully differentiated and males are fertile) and at 3, 6−7 and 10 months (adults). Animal experiments were approved by the Finnish Animal Ethics Committee, and fully met the requirements of the U.S. National Institutes of Health guidelines on animal experimentation.

**Fertility tests**. To assess the ArKI mice fertility, eight ArKI and eight WT male mice at age of 3 months were each mated with two adult WT C57BL/6N female mice. Males were individually placed with WT females and the appearance of vaginal plug was recorded every morning. After 2 weeks, a fresh female was placed with the male. The litter size and sex of pups were recorded from all the females.

**Sperm motility and morphology**. For analyzing sperm motility, sperm from 3-month-old (6 ArKI and 6 WT) mice was collected by making incisions in cauda epididymidis and incubating the tissue in KSOM + AA medium (MR-121-D-UC, Merck, USA) at 37 °C for 15 min. Thereafter, sperm motility was analyzed under a microscope at ×400 magnification. From each sample, the motility of 100 intact spermatozoa (i.e. flagellum attached to head) was classified as progressive, non-progressive or immotile. The average of two analyses from each sample was used for statistical comparison. For the analysis of sperm morphology, spermatozoa released from cauda epididymidis were dispersed on a slide, air-dried at room temperature (RT) and stained with hematoxylin.

**In vitro fertilization (IVF)**. In vitro fertilization was carried out using sperm from 2-month-old male mice. Sperm from six ArKI and six WT mice were collected in K-RVFE-50 medium (CookMedical, Australia) and incubated for 37 °C for 15 min to allow sperm to swim out. IVF was performed by using 50,000 spermatozoa of each mice, incubated in 200 µl of K-RVFE-50 medium, together with oocytes in cumulus oophorus collected from 2 to 4 euthanized, superovulated FVB/N females. The zygotes were washed through three drops of K-RVFE-50 after 6-h incubation, and further incubated overnight. The number of fertilized oocytes was determined the following day as the percentage of oocytes cleaved to two- cell-stage embryos.

**Acrosome reaction**. To induce the acrosome reaction with solubilized zonae pellucidae, female mice were superovulated by Ultra-superovulation using 0.1 ml inhibin antiserum and 3.75 IU equine chorionic gonadotropin (eCG; CARD HyperOva, Cosmo Bio, Japan) combined with 7.5 IU human chorionic gonadotropin hCG[32]. Cumulus-enclosed oocytes were collected from the oviducts in M2 medium (MR-015-D, Merk) containing 0.6 mg/ml hyaluronidase (Sigma-Aldrich, USA) to remove cumulus cells. Cumulus-free oocytes were then transferred into fresh M2 medium and washed four times by transferring into new drop of M2 medium. Zona pellucidae and oocytes were separated mechanically by pipetting up and down with pasteur pipettes with narrow opening. The oocyte-free zona pellucidae were counted and transferred into fresh medium and heat-solubilized for 5 min at 60 °C. For analysis of acrosomal exocytosis, caudal sperm from 3-month-old ArKI and control mice ($n = 4$ for each genotype) were collected and allowed to capacitate for 75 min at 37 °C in KSOM medium. As a control, sperm were incubated in bovine serum albumin (BSA) buffer only (noncapacitated). Acrosome reaction was induced by incubating $1 \times 10^6$ sperm with 50 heat-solubilized mouse zona pellucida or with 10 µM calcium ionophore (A23187, Sigma-Aldrich) for 15 min at 37 °C. As vehicle control, sperm were incubated in 0.5% ethanol (EtOH). The sperm acrosome region was visualized by staining in 0.22% Coomassie brilliant blue (Sigma-Aldrich) in 50% methanol and 10% acetic acid for 2 min. For each genotype, 100−200 spermatozoa from each mouse were classified as either acrosome intact (a bright blue staining on the dorsal region of the acrosome) or with broken acrosome region (patchy or absent staining). Acrosome-reacted sperm were counted and normalized to noncapacitated sperm.

**Histological analysis**. For histological analysis organs were collected from all the above-mentioned age groups. Testis, epididymis, prostate and other organs were fixed in formalin or Bouin's fixative (testis) at room temperature for 1−2 days, dehydrated and embedded in paraffin. Paraffin blocks were sectioned at 5 µm thickness and stained with hematoxylin and eosin or Periodic acid-Schiff.

**Squash and drying down preparations from the testis**. The morphological analysis of differentiating germ cells at each stage of the seminiferous epithelial cycle was performed by phase contrast microscopy (Leica DMRBE microscope) of stage-specific squash preparations that were prepared as described[33]. For drying down preparations, stage-specific segments of seminiferous tubules were cut and cells were released in 100 mM sucrose. Cell suspension was spread on an object slide with fixing solution (1% paraformaldehyde, PFA, 0.15% Triton X-100, pH 9.0). After the slides were incubated in humidified box at RT overnight, washed with 0.4% photoflo (Kodak) and air-dried, they were used for immunofluorescence stainings.

**Immunofluorescence stainings**. The epididymal sections were rehydrated and antigen retrieval was performed in a pressure cooker for 2 h in citrate buffer (pH 6.0). Blocking against nonspecific binding sites was done with 3% BSA in phosphate buffered saline (PBS) -0.1% Triton X-100 (Sigma-Aldrich) for 10 min at RT. Primary antibody incubations with anti-Keratin 5 (1:100, cat. # RM-2106, Thermo Fisher Scientific), or anti-AR (1:1000, cat. # sc-816, Santa Cruz) were carried out at 4 °C for overnight in the blocking solution. The samples were then incubated with AlexaFluor594 secondary antibody (1:200, cat. # A11037, Thermo Fisher Scientific) in blocking solution for 30 min at RT. Staining for alpha smooth muscle Actin was performed using AlexaFluor 488 conjugated anti-alpha SMA antibody (1:1000, cat. # ab184675, Abcam). After washing, all the sections were mounted with Ultra-Cruz™ Mounting medium containing 4''.6-diamidino-2-phenylindole (DAPI) for nuclear staining (cat. # sc-24941, Santa Cruz).

The drying down preparations of stage-specific testicular cells were post-fixed in 4% paraformaldehyde, and treated with 100 mM ammonium chloride for 2 min and with 0.2% Triton X-100 for 5 min. The slides were blocked in 3% BSA, 10% normal goat serum (NGS) in PBS with 0.05% Triton X-100 for 1 h at RT, followed by primary antibody incubation with an antibody against tubulin alpha (1:1000, cat. # MS-581.P, Thermo Fisher Scientific) for 2 h at RT. After washing with PBS the slides were incubated with a secondary antibody attached to AlexaFluor488 (1:750, cat. # A11029, Thermo Fisher Scientific) and Rhodamine-conjugated peanut agglutinin (PNA, 1:4000, cat. # RL-1072, Vector Laboratories) for 1 h at 37 °C. After washing, the slides were embedded in Vectashield Hardset mounting medium containing DAPI (Vector Laboratories). Leica DMRBE (Carl Zeiss AG) microscope was used for image capturing.

The testis sections were rehydrated and antigen retrieval was performed in a pressure cooker for 2 h in citrate buffer (pH 6.0). Blocking against nonspecific binding sites was done with 10% BSA in PBS-0.1% Triton X-100 for 30 min at RT. Primary antibody incubations with either anti-SMA (dilution 1:1000, cat. # ab184675, Abcam), anti-AR (dilution 1:1000, cat. # sc-816, Santa Cruz) or anti-ESPIN (dilution 1:400, cat. # 611656, BD Transduction Laboratories) were carried out at 4 °C for 2 h or overnight in the blocking solution. After washes, the samples were incubated with AlexaFluor594 or AlexaFluor488 secondary antibodies (1:750, Thermo Fisher Scientific) and Rhodamine-conjugated peanut agglutinin (PNA, 1:4000, cat. # RL-1072, Vector Laboratories) in blocking solution for 1 h at RT. Nuclei were stained with DAPI (Sigma-Aldrich), and the slides were mounted in Prolong® Diamond Antifade mountant (Thermo Fisher Scientific). The imaging was performed with Leica DMRBE (Carl Zeiss AG) microscope.

**Gonadotropin and steroid hormone measurements**. For gonadotropin measurements, blood samples were collected from eight ArKI and eight WT mice at time of sacrifice at the age of 3 months and kept at 4 °C for 15 h. Serum was separated by centrifugation and stored at −20 °C until the luteinizing hormone (LH) and follicle-stimulating hormone (FSH) measurements. Concentrations of LH and FSH were measured by time-resolved immunofluorometric assays as previously described[34,35]. For intratesticular and intraepidymal steroid measurements, tissue samples from ten ArKI and nine WT mice were homogenized in sterile deionized water 1:10 (w/v) using an Ultra-Turrax homogenizer (IKA-Werke, Staufen im Breisgau, Germany; Wilmington, NC). Concentrations of androstenedione (A-dione), testosterone (T), and 5α-dihydrotestosterone (DHT) were measured in testis and epididymis homogenates as well as serum samples using a liquid chromatography–tandem mass spectrometry (LC-MS/MS) method as described previously[36]. Quantification limits for A-dione, T, and DHT were 12, 8, 2.5 pg/ml, respectively.

**RNA isolation and gene expression profiling**. For RNA-seq, IS and caput epididymides were collected from five ArKI and five WT mice and testes from three ArKI and three WT mice at the age of 3 months. Total RNA was isolated from the tissues by using Trisure reagent (Bioline, USA) according to the manufacturer's instruction. The quality of RNA was determined by spectrophotometry and RNA analyzer. RNA samples were processed at the Finnish Functional Genomics Centre at the Turku Centre for Biotechnology using Illumina TruSeq Stranded mRNA Sample Preparation Kit and sequenced with HiSeq 3000 system (Illumina, USA) using 50 bp read length and single end sequencing chemistry. IS and caput epididymidis samples were included in one run and testis samples in another run. The quality of the sequenced reads was checked using FastQC tool version 0.11.4[37]. Star v2.5.2b[38] was used to align the reads to the mouse reference genome mm10, available at UCSC (downloaded from Illumina iGenomes website). The number of

uniquely mapped reads associated with each gene according to RefSeq gene annotation was counted using subreads package[39]. The downstream analysis of the data was performed using R version 3.3[40] and its corresponding Bioconductor module 3.3[41]. The count data were normalized for library size using the Trimmed Mean of M-values (TMM) method implemented in edgeR package[42]. For statistical testing the data were further transformed using the voom approach in the limma package[43]. Differential expression analysis was carried out using the ROTS package[44], which optimizes reproducibility among a family of modified t statistics. The significant genes for pairwise comparison of IS and caput epididymides of ArKI and WT mice were reported using Benjamini−Hochberg adjusted p value (FDR) below 0.01 and absolute fold-change (FC) above two, whereas the significant genes for pairwise comparison of testis samples were reported using FDR below 0.2 and absolute FC above two. Enrichment analysis of the differentially expressed genes was performed with topGO, GOstats and GO.db R/Bioconductor packages[45–47]. The hierarchical clustering of the scaled normalized RPKM expression values of AR-regulated epididymal genes[48,49] significant in the ArKI versus WT comparison (FDR < 0.05 and FC > 2) was performed using Euclidean distance and the average method, implemented in the R package pheatmap.

**RT-qPCR**. The RNA samples from six ArKI mice and five WT mice of 3-month-old, three ArKI and WT samples from 5-week-old as well as biological triplicate samples pooled from two animals from 3- and 2-week-old mice were treated with DNase using DNase Amplification Grade Kit (Invitrogen™, Life Technologies) and 0.5 μg of RNA was used for cDNA synthesis using SensiFAST cDNA synthesis kit (Bioline). The cDNA samples were then used for quantitative PCR (qPCR) reactions. All samples were run in triplicate reactions. The RT-qPCR analyses were carried out from initial segment (IS) and caput epididymides and testis for the following genes: *Ar*, serine peptidase inhibitor, Kazal type 2 (*Spink2*), -type 4 (*Spink4*), -type 5 (*Spink5*), - type 11 (*Spink11*), - type 12 (*Spink12*), - type 13 (*Spink13*) and reproductive homeobox 5 (*Rhox5*). The CFX96 real-time PCR detection system (Bio-Rad) and SYBR Green (Finnzymes; Thermo Fischer Scientific) were used for analyses. The results were normalized to ribosomal protein L19 (*L19*) and peptidylpropyl isomerase A (*Ppia*) expression. The sequences of primer pairs used are in Supplementary Table 5.

**ChIP-sequencing**. The caput epididymidis containing IS from 3-month-old WT and ArKI mice were pooled from three mice for ChIP-seq analysis[49]. Briefly, minced tissues were cross-linked in 1% formaldehyde (Sigma-Aldrich) at room temperature for 20 min. After washing twice with ice-cold PBS, tissues were homogenized in hexylene glycol buffer and sonication using micro-tip sonicator (Sonopuls HD 4050, Bandelin, Germany) was performed in RIPA buffer to yield chromatin fragments of 180–220 bp. Immunoprecipitation was carried out with polyclonal anti-AR antibody[50] (5 μl/IP reaction). The analyses were performed from two biological replicates. ChIP-seq libraries were prepared using NEBNext Ultra II kit (New England Biolabs) and sequenced with HiSeq 2500 at EMBL GeneCore (Heidelberg, Germany). Sequenced raw reads were quality controlled using FastQC and quality filtered using FASTX-toolkit before reads were mapped to the mouse reference genome (mm10). Peaks were called using HOMER software[51] keeping only binding events that occurred in both biological replicates. Furthermore, the sequenced samples had >10 million uniquely mapped reads and all the selected peaks had high similarity between replicates. Signal matrixes for heat map were done in HOMER and visualized using ImageJ. The enriched binding events were grouped by</>2-fold signal-density. DNA motif discovery was done with findMotifsGenome program of the HOMER package. For visualization, the data were converted to TDF format in the Integrative Genomics Viewer (IGV).

**Immunoblotting and immunoprecipitation**. For immunoblotting, IS and caput epididymidis and testis were homogenized in SDS-PAGE sample buffer using Tissuelyser II (Qiagen). Proteins were separated on 7.5% SDS-PAGE and immunoblot analyses were performed using rabbit anti-AR[50] (1:10,000) or anti-β-tubulin (1:5000, cat. # sc-5286, Santa Cruz) with horseradish peroxidase-conjugated secondary antibodies (anti-rabbit IgG, cat. # G21234; anti-mouse IgG, cat. # G21040, Invitrogen) and chemiluminescence detection reagents from Pierce. Representative samples of uncropped blots are presented in the Supplemental Figure 6. For immunoprecipitation, caput epididymides (containing IS) and testes were collected at different age points; 2, 3, 5, 7 weeks of age and adult 3 months of age. In adults, four epididymides and two testes from adult WT or ArKI mice were pooled into one sample, whereas from younger animals 4−30 caput and 2−8 testes were pooled. Samples were homogenized in isotonic nondenaturing lysis buffer (150 mM NaCl, 5 mM ethylenediaminetetraacetic acid (EDTA), 50 mM Tris-HCl pH 8.0, 1% Triton X-100, 1× complete mini mix (Roche), 0.2 mM phenylmethylsulfonyl fluoride (PMSF), 1 mM dithiothreitol (DTT) and 20 mM N-ethylmaleimide). Lysates were centrifuged 10 min, 14,000 × g at 4 °C and supernatants precleared using Dynabead Protein G (Thermo Fisher Scientific). Rabbit anti-AR[50] or rabbit IgG (Neomarkers, NC-100-P) was added to lysates and kept in rotation for 2 h 4 °C before immunoprecipitation using Dynabead Protein G. Immunoprecipitates were resolved by SDS-PAGE and immunoblot analyses were carried out using anti-SUMO-1 (1:500, cat # 33-2400, Zymed) or anti-SUMO-2/3 (1:1000, cat. # M114-3, MBL) with horseradish peroxidase-conjugated secondary

antibodies (anti-mouse IgG, cat. # NA931; anti-rabbit IgG, cat. # NA934) with chemiluminescence detection reagents from GE Healthcare Life Sciences.

**Statistical analyses**. For statistical analyses GraphPad Prism 7.0 software (GraphPad Software, USA) was used. The normality of the data was determined by D'Agostino–Pearson and Shapiro–Wilk normality tests. The statistical difference between the two groups was determined by two-tailed Student's $t$ test or by Mann−Whitney test for normally and not normally distributed data, respectively. $p \leq 0.05$ was assigned as the limit of statistical significance. All results are shown as boxplots where center lines indicate the median and the boxes extend to the lower and upper quartiles with whiskers depicting maximum and minimum. The statistical analysis of integrated AR ChIP-seq peaks associated with closest gene and differentially regulated genes in RNA-seq data was performed using Fisher's test in the R package.

**Reporting Summary**. Further information on experimental design is available in the Nature Research Reporting Summary linked to this Article.

## Data availability

RNA-seq and ChIP-seq data generated in this study were deposited in the GEO database under accession number GSE121152. The data that support the findings of this study are available from the corresponding author upon reasonable request.

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

## Acknowledgements
The ArKI mouse line was generated in collaboration with the Turku Center for Disease Modeling, TCDM, Turku, Finland (www.tcdm.fi), and the authors thank the TCDM personnel, N. Messner, H. Niittymäki, K. Hovirinta, J. Palmu, M. Niiranen, T. Kirjonen, and Erica Nyman for skillful assistance in various stages of this study. They would further like to thank J. Järvi, J. Airaksinen and M. Räsänen for technical assistance. This work was supported by the Cancer Society of Finland, the Sigrid Jusélius Foundation and the Academy of Finland.

## Author contributions
F.-P.Z., N.K., M.P., P.S. and J.J.P. designed the main line of the research and carried out majority of the studies, with the essential contribution by M.M., T.L., T.J. and H.K. (molecular biology analyses), A.M., A.L. and L.L.E. and E.A.N. (RNA- and ChIP-seq data analyses), C.O. (steroid biology and analyses); F.-P.Z., N.K., P.S. and J.J.P. wrote the first draft of the manuscript, edited by all the others.

## Additional information

**Competing interests:** The authors declare no competing interests.

