## [Peer Review File · Nature Communications]

Reviewers' comments:

Reviewer #1 (Remarks to the Author):

The authors have produced mutant mice in which two sumoylation sites in the androgen receptor (AR) have been mutated (lysine to arginine substitutions). They demonstrate that male mice are infertile due to very specific dysfunctions of the epididymis. RNA-seq analysis revealed misregulation of genes involved in sperm maturation and function in epididymal tissues, but not in the testis. Consistent with the tissue specific effects, IP-western blot analysis revealed that the AR was specifically modified by SUMO2/3 in the epididymis but not the testis of wild type mice.

Previous work has demonstrated that sumoylation regulates AR activity in cultured cells, but its function in vivo has remained unclear. The work presented here provides evidence that sumoylation is critical for AR-regulated gene expression in a precise, tissue specific manner. The effects of sumoylation on tissue-specific gene expression in a whole organism is relatively unexplored, thus the findings are interesting and important. Overall, the manuscript is well written and the data justify the conclusions.

Reviewer #2 (Remarks to the Author):

The study by Zhang et al, investigates the in vivo functional roles of AR sumoylation after generating a knock-in mouse model that eliminates sumoylation at the two major AR sumoylation consensus sites. Extending their previous in vitro studies, they report that loss of AR sumoylation affects male fertility by interfering with sperm maturation and motility. Transcriptome analyses of mutant testicular and epididymal tissues confirm that the top differentially expressed genes are associated with spermatogenesis, consistent with the observed infertility phenotype. While it is not unexpected that AR sumoylation would impact male fertility, it is of interest that the effects of AR sumoylation on sperm maturation is stage- and tissue-specific, affecting AR function primarily in the epididymis rather than Sertoli/Leydig cells. As with other nuclear receptor sumoylation, AR expression is not affected in the SUMO mutant. Potential impact for male infertility is of interest as pointed out at the end of the discussion.

One of their main findings is the fact that the extent of AR sumoylation appears to be much greater in the epididymis versus the testis. To be more convincing, and given that this assertion is one of the main points of the study, one would like to see this repeated in at least 3 independent biological replicates. (one would also like to see less cropping of Western blots).

In its current form, the study is of interest but lacks mechanistic insight as to which gene pathways in the epididymis are responsible for the observed phenotype. Follow up in vitro studies might improve upon this point.

Minor Points:

Some of the data presented in Table 1 are quite impressive, the impact of these data might improve if plotted as bar graphs showing mutant values compared to values for wild type allele.

Figure 3 – High magnification and better labeling of panels a and b would be helpful in making differences clearer – especially for panels in b.

Parts of Supplemental Fig 3 should be moved into main text.

Figure 5 – Venn diagram is not particularly instructive and could be improved.
Figure 5 – Insertion of ND in bar graphs would help with interpretation.

Figures could easily be condensed into 3-4 figures.

Reviewer #3 (Remarks to the Author):

The authors performed a relatively comprehensive analysis of a SUMOylation deficient AR knockin mouse model with results indicating a defect in the epididymis underlies a defect in sperm and infertility observed in this mouse model. Replacement of 2 SUMOylation sites by arginines triggered male infertility problems, with no apparent phenotype in androgeno-dependent organs from week 3 to week 10 of age, except for the epididymis.

Several AR knock-out mouse models (either conditional or general) have previously demonstrated the importance of androgen signaling in male fertility control. In addition, AR SUMOylation has already been described in 2000 as a mechanism of gene expression regulation. The novelty of this study thus solely resides in the fact that an "epididymis-specific" AR SUMOylation process seems to control of post-testicular sperm maturation event important to male fertility. While these findings are important, notably in terms of male contraceptive development, further mechanistic emphasis should be given to explain this epididymis-specific response.

Several issues exist with the study and manuscript as it stands that need addressing:

1- In several places in the manuscript, the authors need to be careful with their wording so as to not overstate the conclusions from their findings. For example, in the introduction they state "Detailed analysis of the mutant mice revealed, that the conserved SUMOylation sites of AR that show SUMO-2/3 modification in wild-type (WT) epididymis but not in testis, are essential for the proper gene expression in the epididymis and male fertility." Although the study indicates this, the findings still do not preclude the possibility that sperm exiting the testes are not normal as further studies (epididymis specific mouse models or fertility analysis of wild type animals receiving spermatogonial stem cells from KI animals) would be required to test this. Indeed the possibility that sperm exiting the testes are not normal should be mentioned in the discussion and the authors should remain cautious in their wording.

2-The authors show that modification of AR by SUMO-2/3 (but not by SUMO-1) appears to occur exclusively in WT epididymis but not in WT testis. However, the authors do not mention at what time point their analysis was done, and it is possible that SUMOylation of AR in the testes may be occurring at earlier critical time points than the one they chose to analyze. The authors should include additional earlier time points (particularly prior to puberty) in their analysis beginning preferably with the earliest time point AR is expressed in these organs.

3-After determining when AR is first SUMOylated in testis and epididymis, to be able to determine genes directly regulated by AR SUMOylation, the authors should perform gene expression analysis (RNAseq or otherwise) of testis and epididymis at time points 1-2 days just after AR is first SUMOylated in these organs. Problematically, again, the authors again do not mention at what time point they performed RNAseq analysis of testis and epididymis. The age chosen for these analyses are critical for interpreting the results as a large number of functional consequences (in the epididymis) may have already taken place causing gene expression changes not indicative of a direct effect of the inability of AR to be SUMOylated.

4-Further, if abnormal fertility and sperm function is due entirely to a defect in the epididymis caused by a lack of AR SUMOylation, the authors do not confirm experimentally what this defect is (the downstream mechanism remains unknown as it stands in the manuscript), and only a list of

genes aberrantly altered (but that may in fact not be directly regulated) is presented. Therefore, the paper as it stands remains descriptive (with potentially serious issues relating to the time points used for several analyses) and is not mechanistic. To address this, besides correcting any time point issues, the authors should confirm through additional assays that one or more novel genes found altered in vivo are directly regulated by AR SUMOylation.

Point-by-Point Response to Reviewers

Reviewer 1

We thank the reviewer for his/her positive comments and finding our study interesting and well written and performed.

Reviewer 2

We thank the reviewer for finding our study interesting and his/her valuable comments.

Specific comments/questions:

One of their main findings is the fact that the extent of AR sumoylation appears to be much greater in the epididymis versus the testis. To be more convincing, and given that this assertion is one of the main points of the study, one would like to see this repeated in at least 3 independent biological replicates. (one would also like to see less cropping of Western blots).

Authors' response: We have now repeated the AR immunoprecipitation followed by immunoblotting with antibodies against SUMO-1 and SUMO-2/3 and have now three biological replicates with comparable results. This information has now been added to the manuscript. In addition, we have performed same analysis from younger animals at 2, 3, 5 and 7 weeks of age (requested by reviewer 3) and obtained similar results as from adult animals. The time point data have now been included to revised Fig. 5. Full western blot membranes are provided for the reviewer's perusal.

In its current form, the study is of interest but lacks mechanistic insight as to which gene pathways in the epididymis are responsible for the observed phenotype. Follow up in vitro studies might improve upon this point.

Authors' response: Unfortunately, the available immortalized mouse epididymal epithelial cell line mE-Cap18 cells was not amenable for *in vitro* studies. Instead, we performed ChIP-seq analyses from WT and ArKI epididymes in order to obtain unbiased information about the role of SUMOylation on the chromatin binding of AR. Our genome-wide AR chromatin-binding analyses of ArKI vs. WT mice interestingly revealed that the SUMOylation restricts the AR's binding to chromatin in the epididymis *in vivo*. These data are now presented in new panel e of Fig. 4. The perturbed chromatin binding is likely to reflect altered interactions of the AR mutant with chromatin proteins and other transcription factors and thereby contribute to the dysregulated androgen regulation of the epididymis of the ArKI mice.

Minor points:

Some of the data presented in Table 1 are quite impressive, the impact of these data might improve if plotted as bar graphs showing mutant values compared to values for wild type allele.

Authors' response: The information from Table 1 has been now moved into panels f-h of Fig. 3 and shown as bar graphs as suggested by the reviewer.

Figure 3 – High magnification and better labeling of panels a and b would be helpful in making differences clearer – especially for panels in b.

Parts of Supplemental Fig 3 should be moved into main text.

Authors' response: Higher magnification figures with better labeling in Fig. 3 have now been provided as suggested by the reviewer. In the panel b of Fig. 3, a high magnification of Keratin staining is provided to render the difference in the shape of basal cells clearer for the reader (now panel e of Fig. 3). Supplemental Fig. S3 has now been moved into main manuscript (new panel b of Fig. 3).

Figure 5 – Venn diagram is not particularly instructive and could be improved.

Figure 5 – Insertion of ND in bar graphs would help with interpretation.

Authors' response: Venn diagram has now been removed, as the same information is available in the text of the results section. ND has been added to the bar graphs as suggested.

Figures could easily be condensed into 3-4 figures.

Authors' response: Figures have been condensed by moving the Fig. 4 to the Fig. 2 and supplemental Fig. 3 as a part of Fig. 3. As more data have been added to manuscript and figures, no further condensing has not been done, and the current number of figures is 5 plus 4 supplemental ones.

Reviewer 3

We thank the reviewer for finding our results important and his/her valuable comments.

Specific comments/questions:

1- In several places in the manuscript, the authors need to be careful with their wording so as to not overstate the conclusions from their findings. For example, in the introduction they state "Detailed analysis of the mutant mice revealed, that the conserved SUMOylation sites of AR that show SUMO-2/3 modification in wild-type (WT) epididymis but not in testis, are essential for the proper gene expression in the epididymis and male fertility." Although the study indicates this, the findings still do not preclude the possibility that sperm exiting the testes are not normal as further studies (epididymis specific mouse models or fertility analysis of wild type animals receiving spermatogonial stem cells from KI animals) would be required to test this. Indeed the possibility that sperm exiting the testes are not normal should be mentioned in the discussion and the authors should remain cautious in their wording.

Authors' response: We thank the reviewer for this comment. Unfortunately, generation of an epididymis-specific AR SUMOylation mutant mouse along its analyses is not feasible during the time line of this study. Similarly, spermatogonial stem cell transfer and implantation from ArKI animals to WT mice is technically such a demanding task that it cannot be performed with a sufficiently reliable fashion that would be required for unambiguous interpretation of the results. However, we have toned down our wording at the end of the introduction (1st pg, p. 4) not to overstate our findings. We

also mention in the discussion (1 pg, p. 10) that despite of no apparent testicular phenotype in the ArKI mice, we cannot totally exclude the possibility that their altered testicular gene expression also in part contributes to their compromised sperm function.

2- The authors show that modification of AR by SUMO-2/3 (but not by SUMO-1) appears to occur exclusively in WT epididymis but not in WT testis. However, the authors do not mention at what time point their analysis was done, and it is possible that SUMOylation of AR in the testes may be occurring at earlier critical time points than the one they chose to analyze. The authors should include additional earlier time points (particularly prior to puberty) in their analysis beginning preferably with the earliest time point AR is expressed in these organs.

Authors' response: The original analysis was performed in the adult animals (3 months of age). As suggested by the reviewer, we have now analysed several different developmental time points:

- 2 weeks (14 d); prior to puberty, when the first wave of spermatogenesis in the testis has reached pachytene spermatocyte phase, Sertoli cells are still immature, and epididymal epithelium and segments are undifferentiated.
- 3 weeks (21d); prior to puberty, when early haploid cells have appeared in the testis, and epididymal epithelium and segments start to differentiate.
- 5 weeks (35d); puberty, when the first wave of spermatogenesis is completed, and epididymal differentiation continues.
- 7 weeks (49d); when full spermatogenesis is ongoing in the testis and epididymal epithelium and segments are fully differentiated, and males have reached fertility.

The AR is expressed in the epididymis already during embryonal development and in the testis from d 5 postnatally. Immunoprecipitation from 2-week-old males required pooling 30 caput epididymides in order to have enough starting material. Thus, it is impractical/unethical to collect enough epididymides and testes from much younger animals/embryos. In any case, our analyses now contain two time points prior to puberty. Analysis of younger (2-, 3-, 5-, 7- and 10-week old) mice demonstrated that regardless of hormonal status (pre-puberty, puberty, post-puberty) or differentiation phase of epididymal epithelium, AR is SUMOylated in the epididymis but not in the testis as shown in new panel b of Fig. 5. These data also render it unlikely that there would be major differences in the AR SUMOylation at earlier time points. In addition, if AR SUMOylation would be critical during the same developmental phase of testis development, we would expect to see some histological changes in the testis. Nevertheless, our histological analyses of testes from younger age points did not reveal any differences between WT and ArKI males, further supporting the notion that AR SUMOylation plays a more important role in the epididymal than testicular functions.

3- After determining when AR is first SUMOylated in testis and epididymis, to be able to determine genes directly regulated by AR SUMOylation, the authors should perform gene expression analysis (RNAseq or otherwise) of testis and epididymis at time points 1-2 days just after AR is first SUMOylated in these organs. Problematically, again, the authors again do not mention at what time point they performed RNAseq analysis of testis and epididymis. The age chosen for these analyses are critical for interpreting the results as a large number of functional consequences (in the epididymis) may have already taken place causing gene expression changes not indicative of a direct effect of the inability of AR to be SUMOylated.

Authors' response: As stated above, we could not analyse tissue samples from the earliest point of AR expression. However, our time point series show that in the epididymis, but not in the testis, AR is relatively stably SUMOylated regardless of hormonal status (pre-puberty, puberty, post-puberty) and epididymal differentiation, suggesting that AR is SUMOylated also earlier in the epididymis. The RNA-seq analyses of the testes and epididymes were performed from 3-month old mice. Instead of performing RNA-seq from all earlier time points, we performed RT-qPCR analyses of selected genes in younger animals. Our results provide evidence that that AR target genes *Spink2*, *Spink4*, *Spink5*, *Spink11*, *Spink12* are dysregulated already at the age (3-5 weeks), when we do not yet see histological changes in the epididymis, strongly suggesting that the functional alterations observed in the ArKI epithelium are not secondary to changes in the epididymal epithelium. These data are now shown in Supplemental Fig. S3 as panel b.

4- Further, if abnormal fertility and sperm function is due entirely to a defect in the epididymis caused by a lack of AR SUMOylation, the authors do not confirm experimentally what this defect is (the downstream mechanism remains unknown as it stands in the manuscript), and only a list of genes aberrantly altered (but that may in fact not be directly regulated) is presented. Therefore, the paper as it stands remains descriptive (with potentially serious issues relating to the time points used for several analyses) and is not mechanistic. To address this, besides correcting any time point issues, the authors should confirm through additional assays that one or more novel genes found altered in vivo are directly regulated by AR SUMOylation.

Authors' response: We do not expect the phenotype to be due to changes in the expression of only a single AR target gene, but the defects in AR function are likely to affect to larger set of genes, AR target gene programs. According to our unbiased pathway analyses of RNA-seq-based gene expression data, this is indeed the case, as gene ontology (GO) term analysis revealed a significant enrichment of GO terms related to sperm function and fertilization process in the caput region. Moreover, we have now performed CHIP-seq analysis from the WT and ArKI epididymes *in vivo* in order to obtain unbiased information about the role of SUMOylation on the chromatin binding of AR. Our genome-wide AR chromatin-binding analyses of ArKI vs. WT mice interestingly revealed that the SUMOylation restricts the AR's binding to chromatin in the epididymis *in vivo*. These interesting data are now shown in new panel e of Fig. 4. The perturbed chromatin binding is likely to reflect changed interactions of the AR mutant with chromatin proteins and other transcription factors and thereby contribute to the dysregulated androgen regulation of the epididymis of the ArKI mice.

REVIEWERS' COMMENTS:

Reviewer #2 (Remarks to the Author):

This is a much improved study - the authors were responsive to suggested ways to improve the impact of their work.

Reviewer #3 (Remarks to the Author):

While the manuscript still lacks mechanistic insights, the authors addressed the main concerns previously stated.